# Influenza Vaccination Coverage Rates and Determinants in Greek Children until the Age of Ten (2008–2019), the Rhea Mother–Child Cohort

**DOI:** 10.3390/vaccines11071241

**Published:** 2023-07-14

**Authors:** Marianna Karachaliou, Irene Damianaki, Maria Moudatsaki, Katerina Margetaki, Theano Roumeliotaki, Vicky Bempi, Marina Moudatsaki, Lida Vaia Chatzi, Marina Vafeiadi, Manolis Kogevinas

**Affiliations:** 1Barcelona Institute for Global Health, 08036 Barcelona, Spain; manolis.kogevinas@isglobal.org; 25th Local Healthcare Unit, 71307 Heraklion, Greece; eirinidam1@yahoo.gr; 3Department of Pediatrics, University Hospital of Heraklion, 71500 Heraklion, Greece; mariamoudatsaki@yahoo.gr; 4Department of Social Medicine, Faculty of Medicine, University of Crete, 71500 Heraklion, Greece; katmargetaki@hotmail.com (K.M.); troumeliotaki@gmail.com (T.R.); bempiv@gmail.com (V.B.); marmoudat@hotmail.com (M.M.); bafom@uoc.gr (M.V.); 5Department of Preventive Medicine, Division of Environmental Health, University of Southern California, Los Angeles, CA 90033, USA; chatzi@usc.edu; 6CIBER Epidemiologia y Salud Pública, 28029 Madrid, Spain; 7Campus del Mar, Universitat Pompeu Fabra (UPF), 08003 Barcelona, Spain; 8Hospital del Mar Medical Research Institute, 08003 Barcelona, Spain

**Keywords:** influenza, vaccine, Greece, children, asthma, obesity

## Abstract

Background: In Greece, influenza vaccination is currently recommended for children with high-risk conditions. There are limited data on influenza vaccination uptake among Greek children with and without high-risk conditions. We aim to describe the annual influenza vaccination uptake until the age of ten in a population-based mother–child cohort and identify the factors influencing vaccination rates. Methods: Immunization data from the child’s health cards at 4 and 10 years were available for 830 and 298 children participating in the Rhea cohort (2008–2019). We calculated vaccination coverage by age, winter season and among children with asthma and obesity for whom the vaccine is indicated. Univariable and multivariable stepwise logistic regression models were utilized to identify the association between several sociodemographic, lifestyle and health-related variables and vaccine uptake by age four. Results: By the ages of four and ten, 37% and 40% of the children, respectively, had received at least one influenza vaccination. Only 2% of the children were vaccinated for all winter seasons during their first four years of life. The vaccination rate was highest at the age of two and during the 2009–2010 season. Vaccination rates for children with asthma and obesity were 18.2% and 13.3% at age four and 8.3% and 2.9% at age ten. About 10% of all vaccines were administered after December and 24% of the children received only one dose upon initial vaccination. Children with younger siblings and those who had experienced more respiratory infections were more likely to be vaccinated by the age of four, while children exposed to smoking were less likely to be vaccinated. Conclusions: Children in our study were more likely to be vaccinated against influenza at an early age with the peak occurring at the age of two. Nonetheless, annual vaccination uptake was uncommon. Vaccination rates of children with asthma and obesity were well below the national target of 75% for individuals with chronic conditions. Certain groups may merit increased attention in future vaccination campaigns such as children raised in families with unfavourable health behaviours.

## 1. Introduction

### Background

Influenza virus is one of the most important pathogens, associated with acute lower respiratory infection in young children. The lack of pre-existing immunity and the underdeveloped immune system put children at high risk for severe infection and complications [1,2]. Children are considered to drive influenza transmission, as attack rates are four- to five-fold higher in children compared to adults [3].

Influenza vaccination is an effective and safe way of severe disease prevention, especially in high-risk population groups, including young children less than 5 years old [4]. Vaccination can also reduce transmission and provide indirect protection to vulnerable populations [5]. Despite this strong evidence, influenza vaccination recommendations for children differ significantly among various countries [6]. In the USA, an annual influenza vaccination is recommended for everyone aged 6 months and older. In Europe, only recently some countries, including Spain, Finland, Latvia and the United Kingdom have initiated immunisation programs for healthy infants and young children [7,8]. In Greece, influenza vaccination is recommended for children over 6 months with chronic medical conditions (e.g., pulmonary disease, and obesity (included in 2015)), if they are household contacts of infants <6 months or of people with underlying diseases and for children in closed populations. It is also possible that children without any of the aforementioned indications are vaccinated either due to a doctor’s recommendation or a parent’s willingness. Although when indicated the vaccine is provided free of charge, the cost of the necessary medical visit in order to prescribe and vaccinate the child is not covered if it is in the private sector.

In Greece, there is a lack of data on vaccination rates for the general population as well as for specific risk and target groups. Currently, national annual data on vaccination coverage rates are only available for healthcare workers, with percentages being well below the recommended 75% threshold [9]. For children, the limited number of studies indicate that vaccination rates are quite low, even among high-risk children [10,11,12,13]. However these studies have several limitations: they do not provide data specifically for preschool children, often rely on recruitment from outpatient clinics and rely on parental reports for vaccination. A unique characteristic of influenza vaccination is that it is recommended annually as the vaccine is updated each year according to the major circulating strains. To date, there is a scarcity of studies that examine the vaccination coverage of children in repeated winter seasons particularly during their critical first five years of life when they are at a higher risk of severe disease and complications [14].

The aim of this study is to describe the rates of influenza vaccination uptake across different ages and flu seasons in children with and without medical recommendations, followed up until the age of ten in the Rhea mother–child cohort in Greece. Furthermore, using longitudinally collected data we aim to identify factors influencing vaccination rates.

## 2. Methods

### 2.1. Study Design and Participants

This analysis uses immunization data on 830 singleton children participating at 4 years of age follow-up of the Rhea study. Among them, 298 children had available data up to 10 years of age follow-up. Information was available for almost all children who attended the 4 years of age follow-up (missing for 49 children, 5.5%). The Rhea study is an on-going mother–child cohort that recruited 1606 pregnant women from February 2007 to January 2008 “www.rhea.gr” (assessed on 12 July 2023) and followed up with mothers during pregnancy, delivery, and children up to the present [15]. To be eligible for inclusion in the study, women had to have a good understanding of the Greek language and be older than 16 years of age. The study was approved by the ethical committee of the University Hospital of Heraklion (reference number: 96-6/2/2007) and all participants provided written informed consent.

### 2.2. Influenza Immunization Data

The dates, dosage, and trademark of the child’s influenza vaccination were retrieved from the children’s health cards. In our dataset, some information on the dates of vaccination (n = 4, 0.005%) or vaccine brands (n = 126, 15.2%) was missing. This was either because the information was not recorded or because it was difficult to decipher due to being handwritten. Two trained paediatricians were responsible for the data entry of this information. Data entry errors were identified (e.g., outliers, and impossible values), checked and corrected.

We considered a child as (i) eligible for influenza vaccination during a winter season (1st September to end of April) if he/she was at least 6 months of age during that period and (ii) vaccinated if he/she was vaccinated during that winter season. In turn, we calculated the number of vaccination events per child until 4 and 10 years of age. Vaccination events that occurred in the same winter season were accounted for as one event. We considered the first influenza vaccination as complete when vaccine-naive children aged 6 months through 8 years were administered 2 doses of the vaccine in the same winter season as indicated.

### 2.3. Statistics

Descriptive analyses of the study population characteristics and influenza vaccination uptake at certain ages and winter seasons were conducted. Differences in month of vaccination between children receiving one-dose or two-dose schemes were tested by using a chi-squared test. Univariate and multivariate stepwise logistic regression models based on Wald were fit to determine the association of several variables on vaccine uptake by age four. We selected this age group because they represent a target group for influenza vaccination by World Health Organization (WHO). We considered the following variables as determinants: maternal age at delivery (17–25, 25–35, 35–45 years old); maternal origin (Greek, non Greek); maternal educational level at recruitment (low: ≤9 years, medium: >9 years up to attending postsecondary school education, high: attending university or having a university/technical college degree); breastfeeding (never, ever); child having older siblings at age four (yes, no); child having younger siblings at age four (yes, no); daycare attendance at an early age (<2 years, ≥2 years of age); type of child’s health insurance at the 4th year of life (IKA (Social Insurance Fund), OGA (Farmers’ Social Insurance Fund), OAEE (Fund for Merchants, Manufacturers & Related Occupations)/OPAD (Fund of Civil Servants), other, none; reported by parents/caregivers in the 4th year of life study questionnaire); exposure to passive smoking inside the home at the 1st and 4th year of life (yes, no); area of residency at the 4th year of life (urban, rural); preterm birth (<37 weeks, ≥38 weeks); delivery type (vaginal, caesarean); medical diagnosis of asthma at age four (yes, no; reported by parents/caregivers at study questionnaires); respiratory infections the 1st year of life (0–1, 2–4 infections; including otitis, laryngitis, bronchitis, bronchiolitis and common cold; reported by parents/caregivers in study questionnaires); respiratory infections the 4th year of life (0–1, 2–4 infections; including otitis, laryngitis, bronchitis, pharyngitis; reported by parents/caregivers at study questionnaires); and obesity at age four and ten (yes, no; weight and height measurements at the time of the follow-up were used to calculate child’s body mass index (BMI)) [16]. We evaluated all the determinants in univariate models and also built a final model using a forward stepwise regression procedure including all the described determinants considering 0.1 the significance level for addition to the model and 0.2 for removal from the model. Participants with missing covariates were excluded from the final analysis models. We used medical diagnosis of asthma (reported by parents/caregivers at study questionnaires) and obesity definition (based on the child’s BMI) at the age of 4 and 10 to define children with (i) persistent disease as having the disease of interest at age four and still having the disease at age 10; (ii) transient disease as having at age four the disease but not at age ten or being disease free at age four and only having the disease at age ten and (iii) never diagnosed/defined with the disease. Differences in the number of vaccinations by persistent, transient and never asthma/obesity diagnosis were tested by using a chi-squared test. All tests were 2-sided and we used a *p*-value of less than 0.05 as the level of statistical significance. We performed all statistical analyses using Stata/SE (version 16; StataCorp LLC Texas, USA.).

## 3. Results

### 3.1. Descriptive Characteristics of the Study Population

In this study, 830 singleton children are included with available data from their immunization cards until 4 years of age follow-up (mean, standard deviation (SD) age: 4.2 (0.23) years old). Among them, 298 children had available data up to 10 years of age follow-up (mean (SD) age: 10.9 (0.33) years old). In the total study population, 51% were male, 93.4% were of Greek mothers, 31.6% were of mothers with high educational level, 28.9% resided in rural areas, 11.5% were born preterm, 12.6% were never breastfed and 5.3% had asthma at age four. The vast majority of children were fully up-to-date with routine vaccines (e.g., 99% with DTaP, HiB, HBV, 97% with PVC, and 85% with HAV).

### 3.2. Influenza Vaccination Rate

Until the age of four and ten, 37% and 40% of the children, respectively, had received at least one influenza vaccination (Figure 1). The mean (SD) age at first vaccination was 1.6 (0.5) years old among those who have ever been vaccinated by the age of four. Only 6% of children received their first vaccination after the age of four, among those who have ever been vaccinated by the age of ten.

The majority of ever-vaccinated children did not receive subsequent vaccinations (64% and 57% at the ages of four and ten correspondingly). Moreover, the frequency of annual vaccination was very low and decreased with age. Particularly among those ever vaccinated, 5% (n = 13) were vaccinated for all winter seasons eligible for vaccination (Figure 2). In the overall population, this corresponds to 2% receiving annual vaccination up to the age of four. Up to the age of ten years, no child was vaccinated for all winter seasons.

A peak in vaccination rate by age was observed in the second year of life, with 26.9% of the children being vaccinated at that age (Figure 3). In subsequent years, the vaccination rate steadily declined to 4.7% during the sixth year of life and then remained stable at approximately 2.5% until 10 years of age, based on the smaller population with follow-up data until that age. The highest vaccination rate by winter season occurred during 2009–2010, with 32.9% of children being vaccinated during this season (age range 1–2.3 years old). None of the eligible children was vaccinated during 2007–2008 (age range 6–10 months old).

### 3.3. Month of Vaccination and Number of Doses

Vaccination each season was recorded between late September (2.3%) to April (0.1%) and more frequently occurred during October (37.6%) November (31.5%) or December (19.8%). During the first winter season, the majority of vaccinations occurred by the end of November whereas delayed vaccination was more common in later ages (Appendix A). Upon initial vaccination, a total of 76.1% of children under the age of 4 received the recommended two doses during the same winter season and were classified as fully vaccinated against influenza. Children who first received two doses within their initial vaccination season were more likely to receive the first dose before the end of November compared to those who received only one dose during their first vaccination season (78.9% vs. 54.3%, *p*-value: 0.001). Among all vaccines administered, the majority were of the Vaxigrip trademark (79.1%) (Appendix A).

### 3.4. Influenza Vaccination Rates in Children with Asthma and Obesity

The vaccination rates among children with asthma and obesity were sub-optimal. In detail, 18.2% and 8.3% have been vaccinated against influenza in the previous 12 months among children with a medical diagnosis of asthma at age four and ten, correspondingly. These rates were higher than those among children without asthma (9.5% at the age of four and 1.9% at the age of ten). Among ten children with persistent asthma (medical diagnosis at age four and ten), six (60%) of them have never been vaccinated against influenza (Figure 4a). Vaccination rates among obese children were even lower with 13.3% at age four and 2.9% at age ten being vaccinated the previous 12 months. Even among children persistently obese (n = 11), 73% (n = 8) have never been vaccinated against influenza (Figure 4b). Among children with both asthma and obesity at the age of four and ten, one out of three and one out of six, respectively, were vaccinated against influenza.

### 3.5. Determinants of Vaccination Uptake until Age Four

We aimed to assess the effect of a number of characteristics with ever being vaccinated until age four. Results from univariate and multivariate stepwise logistic regression models are presented in Table 1. Children with younger siblings, and those who have experienced more respiratory infections either in the 1st (OR: 1.81; 95% CI: 1.18–2.77) or 4th year of life (OR: 1.52; 95% CI: 1.03–2.24) were more likely to be ever vaccinated until the age of four. Children exposed to passive smoking were less likely to be ever vaccinated (OR: 0.69; 95% CI: 0.48–0.98). In univariate models, daycare attendance before the age of two was also significantly positively associated with vaccination uptake. The direction of association remained in the multivariable models but was no more significant. In univariable models, less likely to be ever vaccinated were children with “IKA” (OR: 0.53; 95% CI: 0.37–0.74) or “other” (OR: 0.60; 95% CI: 0.37–1.00) health insurance compared to those with “OPAD/OAEE”. Univariate associations for each determinant considered are presented in Appendix A. We repeated the aforementioned analyses among children without asthma or obesity and associations were very similar (Appendix A).

## 4. Discussion

This is the first study in Greece to describe influenza vaccination rates in children, including young children and those with asthma and obesity. Four out of ten children were documented as being vaccinated at least once across childhood and the majority of them did not repeat vaccination. Children were more likely to be vaccinated in their early years with the peak occurring at the age of two. Vaccination rates of children with asthma and obesity were sub-optimal. Certain groups may merit increased attention in future vaccination campaigns such as children raised in families with unfavourable health behaviours (e.g., smoking).

The 2020 UNICEF report for Greece noted that there is inadequate and irregular monitoring and reporting of vaccinations conducted throughout the country [17]. The most recent study at a national level on vaccination coverage was conducted in 2012 among 1042 6-year-old children and identified that 30% of them had received at least one influenza vaccination [12]. We had a slightly higher ever-vaccinated rate in our population. Both our study and the national study in 2012 might be biased either positively or negatively by the 2009 influenza pandemic. For example, in our study, the influenza vaccine was more commonly administered at the age of two which coincided with the winter season of 2009–2010. The utilization of child health care services, including routine visits and visits for other recommended vaccinations, during the first two years of life could have also contributed to the observed high vaccination rates at the age of two. Nonetheless, a similarly high vaccination rate is not seen at the age of one. Additionally, the socioeconomic crisis in Greece starting in 2011 might have negatively impacted the vaccination rates in both studies and might explain the rapid decline in vaccination rates observed in the years following 2011 in our study population. Unfortunately, we did not collect any information regarding reasons for vaccine hesitancy but previous studies have identified socioeconomic factors as major predictors for incomplete and delayed immunizations in Greek children [12,18,19].

No previous data exist on the vaccination rate against influenza among preschool Greek children. WHO lists children under the age of 5 years old as a target group for influenza vaccination. In our study, the highest vaccination rate was observed at the age of two and generally children were more likely to be vaccinated at an early age. Apart from providing a static percentage at each age, we reported on the vaccination uptake during subsequent influenza seasons. This analysis uncovered that only 2% of the children received annual vaccination during their first four years of life leaving the remaining 98% vulnerable to the virus for at least one influenza season. This is particularly concerning given the high risk of severe disease and complications among young children. Thus, we probably need to raise public awareness regarding the importance of annual vaccination in Greek children. Our study also highlights that additional efforts should be made in order to increase the percentage of children who are fully vaccinated (two-dose scheme) upon their initial vaccination and of those who are vaccinated on time at each subsequent flu season.

In our study, the vast majority of children with asthma or obesity were not vaccinated against influenza. Gkenzi et al. reported that 50.6% of children with asthma (n = 178, mean age 7.9 years old (2.5 SD)) attending the Paediatric Respiratory Outpatient Clinic of the University General Hospital of Patras in Greece were vaccinated in the 2018–2019 winter season [11]. It is probable that this vaccination rate is overestimated because it relies on patients with mild–moderate asthma who regularly visit an outpatient clinic, representing a closely monitored population. Additionally, parental reports were used to gather information on vaccination status, which may be more subject to recall bias. We observed a much lower coverage rate at 18.2% for 4-year-old and at 8.3% for 10-year-old children with asthma. This is well below the national target vaccination coverage rate of 75% for individuals with chronic conditions. No previous study reported vaccination coverage for obese children in Greece, which appears to be remarkably low according to our study. This may be explained by the relatively recent (2015) inclusion of obesity as a clinically relevant risk factor in the influenza national vaccination program. In the UK, morbidly obese adults, 16–64 years of age, showed among the lowest vaccination rates against influenza compared to other clinical at-risk groups (e.g., diabetes, and patients with immunosuppression) [20]. Asthma and obesity represent the most common chronic diseases in children in which influenza vaccination is recommended. Nonetheless, the vaccination rates appear alarmingly low among Greek children with other chronic diseases and their families [10,13]. These low rates are probably reflected in admission rates due to influenza in a large paediatric intensive care unit in Greece over a ten-year period in which none of the high-risk children admitted were vaccinated against influenza [21].

The presence of younger siblings appears to positively influence vaccination rates, probably due to the implementation of the cocooning strategy among family members. Children with a previous history of several respiratory infections were more likely to be ever vaccinated at age four. This could be driven by the parent’s willingness to safeguard their children from at least one severe infection or by recommendations of health care providers. In a clinical trial, vaccinated children with recurrent respiratory tract infections had fewer respiratory infections and acute febrile respiratory illnesses, fewer prescribed antibiotics and antipyretics, and missed fewer school days than the controls [22]. It is also likely that children who experience more infections interact more frequently with the healthcare system and probably have higher chances of being vaccinated [23]. Parental smoking was the only determinant negatively associated with vaccine uptake. Several studies have found that unhealthy lifestyles such as smoking and alcohol consumption have a negative impact on influenza vaccine uptake [24].

### Strengths and Limitations

The strengths of this study include its prospective longitudinal design. Information on vaccination is not based on parental reports but on the recorded information in vaccination booklets. This information was extracted from trained paediatricians. It is possible that some influenza vaccine doses were missing from the booklets, which would lead to an underestimate of the number of vaccinated children, but we expect this to be a rare phenomenon based on our experience. Parents were able to provide vaccination booklets at the time of follow-up or send the relevant information by email, photo message or fax. An important strength of our study is that we described repeated vaccination events across the years for each individual. We were also able to describe vaccination rates among children with asthma and obesity that are largely lacking in Greece. We had longitudinally collected information on several potential determinants of vaccine uptake. Results may not be generalizable because vaccination coverage rate and its predictors may vary depending on the population being studied, and the context in which the vaccination is being offered. While the prevailing guidelines have remained mostly unchanged during the study period, except for the inclusion of obese children as a high-risk group in 2015, it is important to consider that vaccination uptake may have undergone changes over time. Factors such as the COVID-19 pandemic could potentially impact current vaccination rates.

## 5. Conclusions

The results of this study fill an important gap of knowledge on the rates of influenza vaccination uptake in Greek children during their first ten years of life. Substantial efforts must be made to improve influenza vaccination rates in children with asthma and obesity as indicated by the national immunization program. It is also essential to promote awareness of the need for annual and on-time vaccination and of a two-dose scheme upon initial vaccination for young children. Finally, we should understand and overcome the barriers to vaccination uptake that exist in specific sub-groups such as in families with already established bad health behaviours (e.g., smoking). These findings together can serve as a benchmark until outputs from the recently established National Child and Adolescent Immunization Registry become available.

## Figures and Tables

**Figure 1 vaccines-11-01241-f001:**
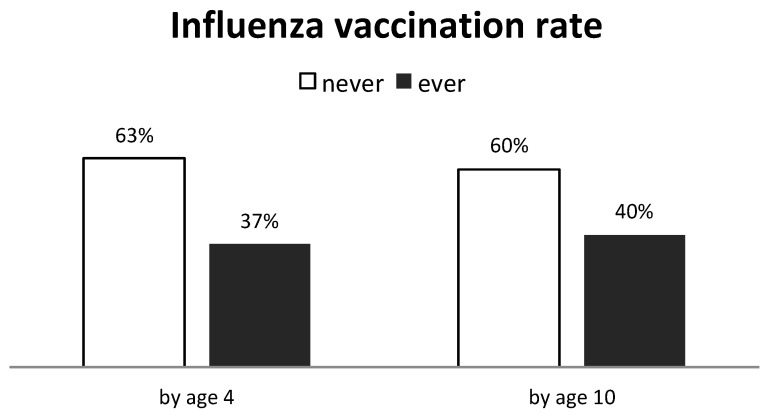
Overall influenza vaccination rate by age four (n = 830) and ten (n = 298).

**Figure 2 vaccines-11-01241-f002:**
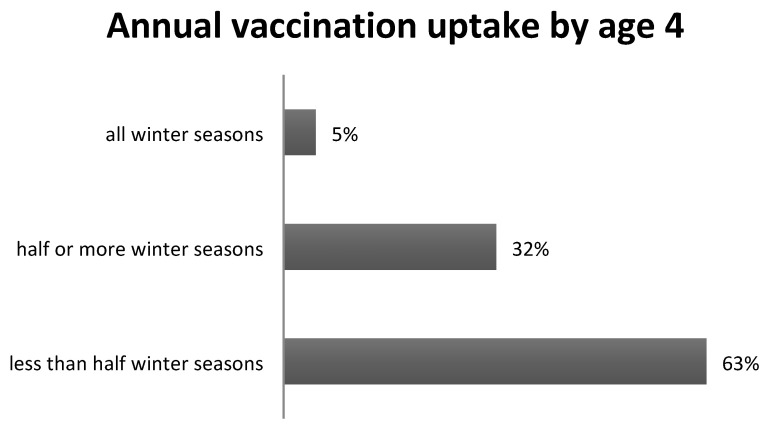
Annual influenza vaccination uptake among those ever vaccinated by the age of four, vaccination by number of winter seasons. Note: Only winter seasons in which children were eligible for vaccination, were considered.

**Figure 3 vaccines-11-01241-f003:**
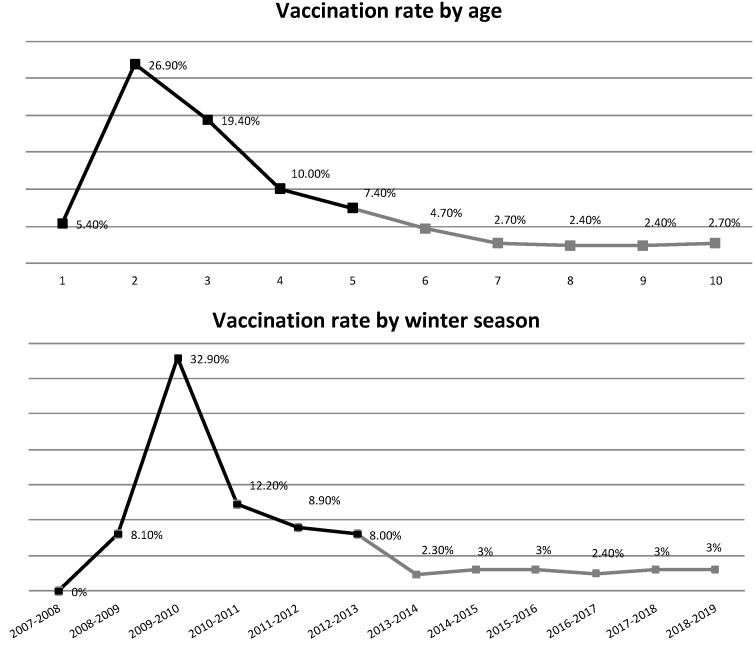
Influenza vaccination rate by age and winter season. Note: Grey colored rates are based on the sub-population of 298 children with data until 10 years of age. The age 11 was excluded from this analysis and line graph, because only a few children were >11 years of age. Similarly the season 2019–2020 was excluded.

**Figure 4 vaccines-11-01241-f004:**
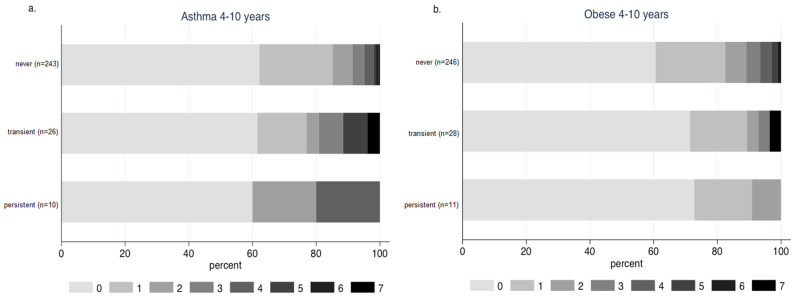
Influenza vaccination events among children with persistent, transient or no (**a**) asthma diagnosis and (**b**) obesity status at the ages of four and ten.

**Table 1 vaccines-11-01241-t001:** Determinants of ever vaccination against influenza by age four selected from stepwise regression models.

	All Children		Children Ever Vaccinated against Influenza by Age Four					
					Univariate Models		Multivariate Stepwise Models ^1^	
	n	%	n	%	OR	95% CI	OR	95% CI
Determinants								
early daycare attendance								
yes	182	22.0	80	26.1	1.44 *	[1.03, 2.02]	1.45	[0.95, 2.20]
no	645	78.0	227	73.9	1.00		1.00	
younger siblings								
yes	317	38.6	137	44.9	1.52 **	[1.14, 2.03]	1.65 **	[1.14, 2.37]
no	504	61.4	168	55.1	1.00		1.00	
passive smoking the first year								
yes	423	56.3	142	51.3	0.72 *	[0.54, 0.97]	0.69 *	[0.48, 0.98]
no	328	43.7	135	48.7	1.00		1.00	
respiratory infections the first year								
2–4 infections	140	21.5	62	26.3	1.54 *	[1.05, 2.25]	1.81 **	[1.18, 2.77]
0–1 infections	511	78.5	174	73.7	1.00		1.00	
respiratory infections the fourth year								
2–4 infections	223	28.1	97	32.9	1.45 *	[1.05, 1.98]	1.52 *	[1.03, 2.24]
0–1 infections	570	71.9	198	67.1	1.00		1.00	

* *p*-value < 0.05, ** *p*-value < 0.01. ^1^ based on 571 children with complete data.

## Data Availability

Data are available upon reasonable request from the corresponding author.

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
