# Peer review of "Influenza Vaccination Coverage Rates and Determinants in Greek Children until the Age of Ten (2008–2019), the Rhea Mother–Child Cohort"

_vaccines, 2023, doi:10.3390/vaccines11071241_

Round 1

Reviewer 1 Report

1.    The authors investigate the influenza vaccination coverage rates and determinants in Greek children by the age four and age ten during 2008-2019, and found that the annual vaccination uptake rate was low and the determinants younger siblings, respiratory infections, or passive smoking may be associated with the vaccination uptake. This study is prospective longitudinal study and the vaccination information is based on the recorded information in vaccination booklets, the conclusion seems to be convincing. However, the conclusion “Children in Greece are more likely to be vaccinated against influenza at an early age with the peak occurring at the age of two” might be biased by the 2009 influenza pandemic. Could the authors give more evidences to exclude or confirm the effect by references comparison?

2.    The figure legend of Figure 4 was not clear.

3.    In table 1, the determinants were based on 571 children with complete data, however, the total numbers in each characteristics of the column “all” were different. Also, the data shown in lane % of the column “ever vaccinated by age four” should be the vaccination numbers/total numbers in lane “all” with same characteristics.     

Author Response

Comment #1.

The authors investigate the influenza vaccination coverage rates and determinants in Greek children by the age four and age ten during 2008-2019, and found that the annual vaccination uptake rate was low and the determinants younger siblings, respiratory infections, or passive smoking may be associated with the vaccination uptake. This study is prospective longitudinal study and the vaccination information is based on the recorded information in vaccination booklets, the conclusion seems to be convincing. However, the conclusion “Children in Greece are more likely to be vaccinated against influenza at an early age with the peak occurring at the age of two” might be biased by the 2009 influenza pandemic. Could the authors give more evidences to exclude or confirm the effect by references comparison?

Response: We thank the reviewer for this comment. We have modified the text in the abstract. In page 9, we discuss how the 2009 influenza pandemic and the socioeconomic crisis starting in 2011 could have impacted the vaccination rates observed in our study.

Comment #2.

The figure legend of Figure 4 was not clear. 

Response: We have modified the legend of figure 4.

Comment #3.

In table 1, the determinants were based on 571 children with complete data, however, the total numbers in each characteristics of the column “all” were different. Also, the data shown in lane % of the column “ever vaccinated by age four” should be the vaccination numbers/total numbers in lane “all” with same characteristics.

Response: We agree with the reviewer. In response, we have made revisions to the table to ensure clarity. The updated table now clearly indicates that the multivariate models were conducted based on a subset of 571 children with complete data. Regarding the additional concern raised by the reviewer, we are uncertain about the specific suggestion put forward. In our study, the results presented under the category of "all" provide the counts and percentages of each determinant within the overall study population. On the other hand, the results presented under the category of "ever vaccinated by age four" display the counts and percentages of each determinant among children who were vaccinated by the age of four. For instance, we found that 26% of children who were vaccinated by age four had attended daycare at an early age, while this percentage was 22% in the overall population.

Reviewer 2 Report

This paper provides a comprehensive look into influenza vaccination rates in children up to ten years old in Greece from 2008 to 2019.The data further exposes surprisingly low vaccination rates even for high-risk groups like children with asthma and obesity. The following changes should be done.

In the introduction mention that Spain has aggregated influenza vaccination for 6 month age  to 59 months https://www.sanidad.gob.es/areas/promocionPrevencion/vacunaciones/programasDeVacunacion/docs/Recomendaciones_vacunacion_gripe_PoblacionInfantil.pdf

Change figure 1,  2,   3there are strange characters.

There are several methods of doing forward stepwise regression (LR, wald etc) you should explicit what method you used

Author Response

This paper provides a comprehensive look into influenza vaccination rates in children up to ten years old in Greece from 2008 to 2019.The data further exposes surprisingly low vaccination rates even for high-risk groups like children with asthma and obesity. The following changes should be done.

Comment #1.

In the introduction mention that Spain has aggregated influenza vaccination for 6 month age to 59 months

https://www.sanidad.gob.es/areas/promocionPrevencion/vacunaciones/programasDeVacunacion/docs/Recomendaciones_vacunacion_gripe_PoblacionInfantil.pdf 

Response: In the modified text, Spain is listed among the countries recommending influenza vaccination in young children. We have included the appropriate reference.

Comment #2.

Change figure 1,  2,   3there are strange characters.

Response: Could you please provide more details or specify the specific strange characters you are referring to? As they are not visible to us, any additional information you can provide would be helpful in understanding the issue.

Comment #3.

There are several methods of doing forward stepwise regression (LR, wald etc) you should explicit what method you used

Response: We have explained in the text that stepwise regression was based on Wald.

Reviewer 3 Report

I am glad to read this paper that r is interesting because it provides a comprehensive look into influenza vaccination rates in children up to ten years old in Greece , reveals important factors influencing these rates, and exposes surprisingly low vaccination rates even for high-risk groups like children with asthma and obesity, raising crucial considerations for future vaccination strategies. Some minor changes has to be done before publication.

·         In the introduction note that Vaccination of children older than six months in Spain is also included in the vaccine schedule. https://www.sanidad.gob.es/areas/promocionPrevencion/vacunaciones/vacunas/ciudadanos/gripe.htm#:~:text=Las%20personas%20mayores%20de%206,el%20primer%20a%C3%B1o%20de%20vacunaci%C3%B3n).

·         There are several ways to do a stepwise regression for example IBM SPSS allows to chose between (LR), Wald, or score/conditional.Please indicate what forward stepwise regression methods you used

 ·         Review figures 1 and 2,  I don`t know what means the  question marks signs “?”  are they  typos?

·         Please introduce the abbreviation the first time they are used in the text, now they are at the end of the manscript.

Author Response

I am glad to read this paper that r is interesting because it provides a comprehensive look into influenza vaccination rates in children up to ten years old in Greece , reveals important factors influencing these rates, and exposes surprisingly low vaccination rates even for high-risk groups like children with asthma and obesity, raising crucial considerations for future vaccination strategies. Some minor changes has to be done before publication

Comment #1.

In the introduction note that Vaccination of children older than six months in Spain is also included in the vaccine schedule.

https://www.sanidad.gob.es/areas/promocionPrevencion/vacunaciones/vacunas/ciudadanos/gripe.htm#:~:text=Las%20personas%20mayores%20de%206,el%20primer%20a%C3%B1o%20de%20vacunaci%C3%B3n).

Response: In the modified text, Spain is listed among the countries recommending influenza vaccination in young children. We have included the appropriate reference.

Comment #2.

There are several ways to do a stepwise regression for example IBM SPSS allows to chose between (LR), Wald, or score/conditional.Please indicate what forward stepwise regression methods you used

Response: We have explained in the text that stepwise regression was based on Wald.

Comment #3

Review figures 1 and 2,  I don`t know what means the  question marks signs “?”  are they  typos?

 Response: Could you please provide more details or specify the specific strange characters you are referring to? As they are not visible to us, any additional information you can provide would be helpful in understanding the issue.

Comment #4

Please introduce the abbreviation the first time they are used in the text, now they are at the end of the manuscript.

Response The abbreviations used in the text are already explained at their first occurrence. We have deleted the list of abbreviations from the end of the manuscript.

Round 2

Reviewer 1 Report

The author have revised the manuscript according to the comments.

Reviewer 2 Report

My decision, is To Accept as it is.
The authors have incorporated all my comments to the manuscript.

Reviewer 3 Report

All issues have been resolved; the only issue is number 3, related to typos in the graphis. I am sending you the PDF that you uploaded. You can see the errors.
